# Toughening and Enhancing Melamine–Urea–Formaldehyde Resin Properties via in situ Polymerization of Dialdehyde Starch and Microphase Separation

**DOI:** 10.3390/polym11071167

**Published:** 2019-07-09

**Authors:** Jianlin Luo, Jieyu Zhang, Qiang Gao, An Mao, Jianzhang Li

**Affiliations:** 1MOE Key Laboratory of Wooden Material Science and Application, Beijing Key Laboratory of Lignocellulosic Chemistry, MOE Engineering Research Centre of Forestry Biomass Materials and Bioenergy, Beijing Forestry University, Beijing 100083, China; 2Collaborative Innovation Center of Sustainable Utilization of Giant Salamander in Guizhou Province, Guizhou Provincial Key Laboratory for Rare Animal and Economic Insects of the Mountainous Region, Guiyang University, Guiyang 550005, China; 3Key Laboratory of State Forestry Administration for Silviculture of the lower Yellow River, College of Forestry, Shandong Agricultural University, Taian 271018, China

**Keywords:** melamine-urea-formaldehyde resin, dialdehyde starch, toughening, in situ polymerization, microphase separation

## Abstract

The goal of this study is to employ bio-based dialdehyde starch (DAS), derived from in situ polymerization and the resultant microphase separation structure, to improve the strength of melamine–urea–formaldehyde (MUF) resin, as well as enhance the properties that affect its adhesive performance. Thus, we evaluated the effects of DAS on the chemical structure, toughness, curing behavior, thermal stability, and micromorphology of the MUF resin. Furthermore, the wet shear strength and formaldehyde emissions of a manufactured, three-layer plywood were also measured. Results indicate that DAS was chemically introduced into the MUF resin by in situ polymerization between the aldehyde group in the DAS and the amino group and hydroxymethyl group in the resin. Essentially, polymerization caused a DAS soft segment to interpenetrate into the rigid MUF resin cross-linked network, and subsequently form a microphase separation structure. By incorporating 3% DAS into the MUF resin, the elongation at break of impregnated paper increased 48.12%, and the wet shear strength of the plywood increased 23.08%. These improvements were possibly due to one or a combination of the following: (1) DAS polymerization increasing the cross-linking density of the cured system; (2) DAS modification accelerating the curing of the MUF resin; and/or (3) the microphase separation structure, induced by DAS polymerization, improving the cured resin’s strength. All the results in this study suggest that the bio-based derivative from in situ polymerization and microphase separation can effectively toughen and enhance the properties that affect adhesive performance in highly cross-linked thermosetting resins.

## 1. Introduction

A urea–formaldehyde (UF) resin is an oligomer obtained by condensation polymerization of urea and formaldehyde. Heating and/or acidification facilitates continuous oligomer polymerization, which develops into an insoluble and infusible cross-linked network that can be used to bond the substrate together and obtain specific mechanical properties. Owing to its fast curing rate, easy adaptability, colorless bond line, excellent wood adhesion capability, and particularly—its low cost— UF resin has been one of the most important adhesives employed in wood-based composite panels manufacturing. However, the formaldehyde–urea reaction is reversible, hence the inevitable existence of free formaldehydes in the resulting resin. In addition, the wood–resin ether bond commonly breaks during use and is a second route for formaldehyde release. Both processes result in harmful formaldehyde emissions from the bonded panels during service life, significantly limiting the indoor application of UF resin. The most effective method of reducing formaldehyde emissions is lowering the formaldehyde-to-urea (F/U) ratio. However, a decrease in the F/U generates a large reduction in the resin’s water resistance —thereby rendering it unusable to the wood panel industry. Thus, studies on UF resin alteration have focused on techniques like copolycondensation modification that are capable of decreasing formaldehyde emissions without decreasing water resistance.

Copolycondensation modification is an effective and commonly used method in which highly reactive compounds, such as melamine or phenol, are introduced into the UF resin to facilitate the development of copolycondensation resins, like MUF and phenol–urea–formaldehyde (PUF) resin. Copolycondensation resins have a proven track record of enhanced performance and lower formaldehyde emissions in comparison to unmodified (i.e., neat) UF resins [1,2]. Owing to its low cost and non-toxic nature, MUF resin is widely used in the preparation of moisture-proof boards. The performance improvement of MUF resin is attributed to melamine’s high reactivity and rigid structure (triazine ring). Because melamine has more functional sites than urea, it reacts with more formaldehyde molecules, which decreases the free formaldehyde concentration in the resin, and subsequently decreases the amount of formaldehyde emissions from the resulting panel. In addition, the presence of more functional sites on the melamine results in a more extensive, stable bond formation, which in turn, increases the branched-type structure and the cross-linking capability of the cured resin. As such, the addition of melamine both reduces the formaldehyde emissions and enhances the resin’s bond strength.

In addition to its tendency toward formaldehyde emissions, UF resin is also significantly problematic due to its inherent brittleness. Because of this characteristic, cured UF resins are very prone to cracking, and therefore exhibit poor impact resistance when applied to the wood panels [3]. Many studies have focused on improving the toughness (i.e., resistance to crack propagation) of UF and MUF resins. These investigations are classified into three primary categories: Reduction of cross-linking density, thermoplastic toughening, and introduction of cellulose-based materials.

With respect to the first category, adding melamine is one option for enhancing the resin. However, since the brittleness of UF resin is related to its high cross-linking density, incorporating melamine provides a more branched, cross-linking network structure, thereby further reducing the brittleness of the resulting resin. Another option was reported in [4,5]. In place of traditional urea, these researchers synthesized resin using di- and tri- functional amines, (e.g., hexamethylenediamine, triethylaminetriamine, triethyleneoxidediamine), which have a lower reactivity than urea and the urea derivatives. The resultant resin exhibited improved toughness, but slowed the curing process and decreased the bonding performance.

Regarding the second category, UF and MUF are typical thermosetting amino resins. Thermoplastic toughening, i.e., toughening thermosets by incorporating thermoplastics to prepare high-performance structural composites, have been widely used in materials science [6,7]. Rachtanapun and Heiden (2003a and 2003b) mixed intermediate hydrophilicity acrylic thermoplastic with UF resin and found that a 5% (*w*/*v*) addition improved the bonded wood flour composite impact strength by 69% via phase separation [8], and further improved the impact strength by 94% by adding 5% (*w*/*v*) acrylic thermoplastic via in situ polymerization. Kim et al., investigated the curing kinetics of in situ modified UF resin after adding different acrylic thermoplastics and explored the relationship between the cross-linking density of the cured resin and water resistance [9]. They found that thermoplastic modification by in situ polymerization decreased the activation energy of the UF resin and that cross-linking density influenced the water resistance of cured thermoplastic-modified UF resin. Vinyl acetate-versatic vinylester copolymer (VA/VeoVA) latex and polyurethane (PU) also exerted significant enhancing and toughening effects on UF resin [10,11]. In a recent study, polyvinyl alcohol was effectively used to improve the toughness of the UF resin foam and lower its thermal conductivity [12]. It has been well demonstrated that in situ polymerization of flexible monomers generates a toughening effect; however, these materials increase the resin’s dependency on petrochemical resources.

Introduction of cellulose-based materials, such as natural fibers and nanocelluloses, define the third category of potential methods for increasing resin toughness. Examples of past successes include grewia optiva, cellulose nanofibrils (CNF), and microfibrillated cellulose (MFC). Singha and Thakur [13], used the grewia optiva fiber to enhance the mechanical properties and improve the flexibility of a UF resin/fiber composite; Veigel and Müller et al., reported that the fracture energy of the bonded double cantilever beam increased by 45% after combining 2 wt % untreated CNF powder with UF resin; CNF also substantially improved the fracture energy and fracture toughness in the UF resin-bonded particle board and oriented strand board; and contributed to an increased toughening effect in the MUF resin [14]. Furthermore, 3 wt % MFC added to UF resin induced a 5.7% increase in the tensile shear strength of wood bonded with that adhesive [15]. Moreover, the effects of MFC addition on UF resin’s thermal properties was recently evaluated, and MFC was found suitable for the modification of UF resin with a low formaldehyde/urea mole ratio [16].

While increasing toughness has been proposed and evaluated in numerous, varied studies, little attention has been paid to the in situ polymerization of bio-based materials for enhancing UF and MUF resin. Ping Li et al. [17] used oxidized starch as a filler to modify UF resin and found it could react with UF polymers. Recently, Ping Li et al. [18] synthesized a resorcinol-dialdehyde starch-formaldehyde copolycondensation resin with comparable bonding properties to costly resorcinol-formaldehyde resin, showing a potential for commercial application. Thus, this work introduces a flexible bio-based polymer that can react with MUF resin molecules, improve the toughness, and enhance the performance of the MUF resin; while increasing the biomass content of the resin system. 

Dialdehyde starch (DAS) is an important derivative of natural starch. Due to its cross-linking ability and biodegradation capability, DAS has the potential for extensive industrial applications. It is often used as a wet strength agent for paper, in plastic packaging material [19], as a biodegradable carrier for drug delivery [20], and as a heavy metal absorbent [21]. DAS is also a flexible bio-based polymeric dialdehyde that can react with alcohols, amines, and acids. Therefore, DAS is a potential modifier for in situ polymerization with MUF resin, while increasing biomass content. In the present study, DAS was introduced into MUF resin to improve toughness and enhance its adhesive properties.

## 2. Experimental Procedure 

### 2.1. Materials

Reagent-grade formaldehyde (37%) and technical-grade urea and melamine (purity 99%) were provided by Tianjin Chemicals Co. Ltd., Tianjin, China. Dialdehyde starch, with an oxidizability of 96%, was prepared from the periodate oxidation of cassawa starch, and supplied by Jinshan Denaturated Starch Co. Ltd., Shandong, China. Aqueous solutions of formic acid (20%) and sodium hydroxide (NaOH) (30%) were used to adjust the pH during the synthesis process. An aqueous solution of ammonium chloride (NH_4_Cl) (20%) was used as a hardener to provide acid curing conditions. Other analytical reagents were provided by Lanyi Chemical Co. Ltd., Beijing, China. 

### 2.2. Synthesis and Preperation

#### 2.2.1. Synthesis of Neat and DAS Modified MUF Resins

The neat MUF resin with 5% melamine (based on the weight of liquid resin) and an F/(U+M) mole ratio of 1.1:1 was prepared as a control, following the traditional, three-step, alkali–acid–alkali synthesis. A 37% formaldehyde solution was poured into a reactor; and the pH was adjusted to 8.0 using a 30% NaOH solution. The first portion of urea (U_1_) and melamine was added to obtain an F/(U_1_ + M) mole ratio of 2:1. The mixture was then heated and maintained at 90 °C for 1 h. Next, 20% formic acid was added to obtain a pH of 6.0 and start the acidic reaction. Condensation progressed until reaching a target point determined by flocculent appearance, after which the mixture was poured into a beaker of 20 °C water. The reaction was stopped by adjusting the pH back to 8.0. The second portion of urea (U_2_) was then added to obtain a final mole ratio of 1:1. The mixture was stirred at 85 °C for 30 min, after which the liquid resin was cooled to ambient temperature and the final pH was adjusted to 8.0.

The DAS-modified MUF resins with the same melamine content and F/(U+M) mole ratio were prepared by adding DAS with the first portion of urea and melamine. The DAS was added in 1%, 3%, 5%, and 7% concentrations relative to the liquid resin weight. 3% DAS was added with the second portion of urea (U_2_) to synthesize a contrast resin such the chemical reaction between DAS and MUF resin was observable.

#### 2.2.2. Preparation of Plywood

Three-ply poplar plywood was prepared at 120 °C hot pressing temperature and 1.0 MPa pressure. The 1% NH_4_Cl solution (based on the liquid resin) was used as the curing agent. The adhesive spread rate was 160 g/m^2^ (single glue line) and the hot-pressing time was 6 min. After hot pressing, the plywood was stored under ambient conditions for at least 24 h before the wet shear strength was tested. Three plywoods were prepared for each resin formulation.

### 2.3. Measurement

#### 2.3.1. Physical Properties of Resin

To measure the resin’s solid content, ~1 g of resin was poured into a disposable aluminum dish, weighed, oven dried for 3 h at 105 ± 2 °C, then weighed again. The solid content was calculated based on the weight of the resin before and after drying. The viscosity at 25 °C was measured using a Brookfield DV-II viscometer (Middleboro, Massachusetts, USA) with a 61# spindle at a spinning rate of 100 rpm. Curing time was measured in accordance with China National Standards (GB/T 14074-2013) [22]. ~50 g of resin and 2 mL of 25% NH_4_Cl solution were added to a beaker and stirred evenly, while 10 g of mixed resin was put in a test tube and immersed in a boiling water bath. The mixed resin was stirred continually with a rod until the rod could not be moved or the resin hardened abruptly. Curing time was measured from the time the test tube was immersed in the water bath to the time the resin exhibited hardening. For all three measurements—solid content, viscosity, and curing time—three replications (error less than 5 s) for each resin were conducted, and the average was reported as the final value.

#### 2.3.2. Fourier Transform Infrared (FTIR) Spectroscopy

The resins were placed in an oven at 120 ± 2 °C until they reached a constant weight, indicating they had completely cured, then ground into fine powder. The powder was mixed with KBr crystals at a mass ratio of 1/100, then pressed into a special mold to form a sample folium. The FTIR spectra were recorded with 32 scans, using a Thermo Nicolet 6700 FT-IR (ThermoFisher, Madison, WI, USA) over the range of 400 to 4000 cm^−1^ with a 4 cm^−1^ resolution.

#### 2.3.3. ^13^C Nuclear Magnetic Resonance (NMR) Spectroscopy

The ^13^C NMR spectra were obtained using a DELTA2 600 NMR spectrometer (JEOL RESONANCE Inc., Tokyo, Japan). The liquid resins were dissolved in deuterated dimethyl sulfur oxide (DMSO-*d*_6_). Samples were scanned 2560 times using a resolution, pulse width, relaxation delay, and repetition time of 1.45 Hz, 12.75 μs (30°), 20, and 20.69 s, respectively. Decoupling was employed to minimize the nuclear Overhauser effect. The spectral values of urea carbonyls, melamine triazine carbonyls, and methylenic carbons were integrated under the same scale factor and quantified as group percentages. Urea carbonyls were separated into their substitution patterns of free urea, mono-substituted urea, di-/tri-substituted urea, and cyclic urea. Melamine triazine carbonyls were separated into free and substituted melamine.

#### 2.3.4. X-ray Diffraction (XRD) Analysis

The cured resin powders were analyzed at ambient temperature by an XRD diffractometer (D8 ADVANCE, BRUKER, German) using a Cu Kα X-ray source with a wavelength (λ) of 1.5405 Å, in the 2θ scan ranging from 5° to 60°. The resin samples’ crystallinity was calculated based on an area method, using EVA V 3.1 software (BRUKER, German) [23].

#### 2.3.5. Differential scanning calorimetry (DSC) analysis

The resin samples were mixed with 1% solid ammonium chloride (based on the weight of liquid resins) and then freeze-dried for 12 h. ~5 mg of the freeze-dried sample was placed in an aluminum pan. The pan was sealed and heated from −10 to 130 °C at a heating rate of 10 °C/min using a TA Instrument (DSC Q2000, Waters Company, New Castle, DE, USA). Each measurement was performed under an N_2_ flow with a 50 mL/min flow rate.

#### 2.3.6. Scanning Electron Microscope (SEM) Analysis

The resins were placed in an oven at 120 ± 2 °C until completely cured. The cured MUF resins were fractured and the cross-sections were evaluated. To begin, the samples were placed on an aluminum stub. Next, a 10 nm Au/Pd film coating was applied using a Q150T S Turbo-Pumped Sputter Coater/Carbon Coater (Quorum Technologies Ltd., East Sussex, UK). The coated samples were then examined and imaged using a JSM-6500F field emission scanning electron microscope (FESEM) (JEOL USA Inc., Peabody, MA, USA).

#### 2.3.7. Mechanical Properties Test

The resin-impregnated paper was prepared as described in [24]. A tensile testing machine (DCP-KZ300) with a loading speed of 50 mm/min and an initial gauge length of 50 mm was used to characterize the mechanical properties of: (1) Paper impregnated with neat MUF resin, and (2) paper impregnated with MUF resin modified with DAS. Stress–strain curves were generated using the average of five measurements for each sample. Tensile strength and elongation at break were determined by calculating the average values. The thickness and width were measured using a digimatic micrometer and a vernier caliper, respectively. Measurements were made three times and the average value reported.

#### 2.3.8. Wet shear Strength of Plywood

The wet shear strength of the plywood was determined in accordance with the China National Standard GB/T 17657-2013 for Type II plywood [25]. The prepared plywood pieces were cut into shear specimens (gluing area of 25 mm × 25 mm) and then submerged in 63 °C water for 3 h. The wet shear strength values of the wood specimens were tested using a common tensile machine operated at a speed of 10.0 mm/min. The force (N) at which the bonded wood specimen got damaged was recorded. The reported strength data of the adhesives were the average of ten replications from two plywood samples. Wet shear strength (MPa) was calculated using the following equation (Equation (1)):
(1)$$\mathrm{Wet}\;\mathrm{Shear}\;\mathrm{Strength}\left(\mathrm{MPa}\right)=\frac{\mathrm{Force}\left(N\right)}{\mathrm{Gluing}\;\mathrm{area}\;\left(m^{2}\right)}$$

#### 2.3.9. Formaldehyde Emission of Plywood

The plywood’s formaldehyde emission was determined using the desiccator method in accordance with the procedure described in the China National Standards (GB/T 17657-2013). After being stored in a ventilated environment for 20 days, 50 mm × 150 mm plywood slabs were prepared. Specimens of each panel were placed in a 9–11 L sealed desiccator at 20 ± 2 °C for 24 h. The emitted formaldehyde was absorbed by 300 mL deionized water in a container. The formaldehyde concentration in the water was measured using a visible spectrophotometer. The formaldehyde concentration was measured in three panels of the same sample and the average was calculated and recorded.

## 3. Results and Discussion

### 3.1. Physical Properties of Resin

Figure 1 shows the physical appearance of the neat MUF and DAS-modified MUF resin samples. Amino resins (UF, MF, and MUF resin) have long been considered colloids [26]. The neat MUF resin presents as a suspension of whitened colloidal particles in a water-dispersed medium. The contrast resin (MUF/3%DAS)—prepared by adding 3% DAS with the second portion of urea and melamine(second stage)—shows a faint yellow tint. This visible difference indicates that the DAS did not participate in the reaction, but instead dissolved in the water; a likely result of the reaction between the aldehyde group in DAS and the amino/hydroxy group being unable to occur due to the alkaline condition in the second stage. Contrarily, the 3%DAS/MUF sample, which was synthesized with DAS accompanying the first portion of urea and melamine, remained white and colloidal, revealing that polymerization between DAS and MUF resin had occurred. In this case, the added DAS underwent acid condensation during resin synthesis and was therefore able to polymerize with the MUF resin. However, the increase in DAS (5% and 7%) rendered the resin transparent. Because DAS is a biomacromolecule, its polymerization with MUF resin increased the length of the molecular chain, affecting the formation of a colloid particle, resulting in a transparent visible appearance.

The basic physical properties of the neat and DAS-modified MUF resins are summarized in Table 1. Both the solid content and viscosity of the MUF resin increased as DAS was added, eventually leading to enlarged solid structures and a highly viscous modified resin. By increasing the solid component via polymerization, introduction of DAS also increased the molecular weight of the resin system. Furthermore, the viscosity of the modified resin increased in response to a large number of hydroxymethyl groups in the DAS molecules forming hydrogen bonds with water and MUF resin molecules. By contrast, curing time, which serves as an indicator of resin reactivity, initially decreased with the addition of DAS. However, the addition of excessive DAS reversed the trend and caused the curing time to increase; thus, 3%DAS-modified MUF resin showed the shortest curing time. This result demonstrates that the modification, with the appropriate amount of DAS, improves the reactivity of the MUF resin; possibly due to an increase in the branched molecular chain derived from DAS polymerization. However, when an excessive amount of DAS was added, the thermoplastic nature of DAS limited the curing reaction of the resin.

### 3.2. Polymerization of DAS and MUF Resin

The quantitative solution ^13^C NMR spectra of neat and DAS-modified MUF resins are presented in Figure 2. The integration percentages of various methylene and carbonyl carbons are summarized in Table 2. The definition and assignment of carbon groups and quantitative analyses were determined in accordance with previous studies. The spectral values of carbon from urea, melamine, and formaldehyde were integrated and then quantified as group percentages. Urea carbon was separated into free urea and their substitution patterns of mono- and di-/tri-substituted urea. The triazine carbons of melamine were divided into free melamine and substituted melamine. The methylene carbons in hydroxymethyl, methylene, and methylene ether groups were obtained from formaldehyde and then independently quantified. The hydroxymethyl groups from hydroxymethylmelamine or hydroxymethylurea could not be differentiated in the spectra; thus, they were counted together for quantitative analysis.

The chemical structures of various carbon groups are presented in Figure 3. Chemical shifts at 47.1, 53.9, and 55.6 ppm are assigned to type I, type II, and type III methylene, respectively. Signals at 69.3, 75.2, and 78.5 ppm belong to type I, type II, and type III methylene ether, respectively. Shifts at 64.3 and 70.6 ppm correspond to type I and type II hydroxymethyl groups, respectively. Type I, type II, and type III methylene or methylene ether vary with respect to the number of substituted hydrogen atoms in -NH- attached to methylene or methylene ether. Similarly, type I and type II hydroxymethyls indicate that the hydrogen atoms of urea amino were substituted by one or two hydroxymethyls. Free formaldehyde slightly shifted at 82.9 ppm. Carbonyl signals of free urea, mono-, di-, and tri-substituted urea were observed at 161.9, 160.3, and 158.9 ppm, respectively. Signals of cyclic urea carbons occurred at 154–155 ppm. Peaks of free and substituted melamine triazine carbons occurred at 167.8 and 166.8 ppm, respectively. A small peak at 49.15 ppm was assigned to methanol, which was used as a stabilizer in the formaldehyde aqueous solution.

As shown in the spectra, all resins exhibited similar chemical structures, with three exceptions that resulted from the incorporation of DAS: (1) The peaks related to urea carbonyl carbons and melamine triazine carbons shifted after the introduction of DAS, revealing a chemical reaction between DAS and urea/melamine. (2) There was a significantly low concentration of methylene groups in the DAS-modified MUF. This phenomenon can be explained by the understanding that DAS consumed urea and melamine. The residual urea and melamine then reacted with formaldehyde to produce more hydroxymethylureas and hydroxymethylmelamines, thereby forming methylene ether linkages in subsequent condensation. (3) Cyclic ureas emerged in the DAS-modified MUF resins. The chemical reaction of DAS and urea/melamine increased the length of the molecular chain. Owing to high steric hindrance, these extended molecules required increased acidity for polymerization. The strongly acidic environment promoted the formation of cyclic ureas. The nature of DAS is similar to that of aldehyde group compounds, which can react with aminos, acids, and alcohols. Urea and melamine are amino compounds, and they exhibit polymerization with DAS. The polymerization of DAS with urea and hydroxymethylurea, respectively, is illustrated in Figure 4. These reactions facilitated the in situ polymerization of DAS with the MUF resin molecule, subsequently forming a new hybrid adhesive system.

Table 2 shows the effect of DAS concentration on the chemical structure of modified MUF resin. Note that the hydroxylmethyl content increased in parallel with the addition of DAS. This increase resulted from hydroxymethylation after the second portion of urea (U_2_) was added. The in situ polymerization of DAS consumed some urea/melamine, but formaldehyde remained, leading to a high concentration of available formaldehyde to react with urea (U_2_). Consequently, more hydroxylmethyl groups were produced in the modified resins with a high DAS concentration; and high hydroxylmethyl content in resin facilitates the curing process. By contrast, the total methylene ether content decreased with an increase in DAS concentration. Methylene ether was formed by the condensation reaction between hydroxymethylureas in the first hydroxymethylation stage and the acidic condensation stage of resin synthesis. The in situ polymerization of DAS and hydroxymethylurea (Figure 4(2)) reduced the formation of methylene ether linkages and also increased the free formaldehyde content in resin.

Table 2 shows important effects on other system constituents, including those of urea carbonyl carbons: (1) Free urea decreased in parallel with the continuous addition of DAS; thereby confirming the in situ polymerization of DAS and urea. (2) The mono-substituted urea noticeably decreased with the introduction of DAS, but then increased as more DAS was added. (3) The di-/tri-substituted urea markedly increased and exhibited the highest concentration in the 3%DAS/MUF resin. The high concentration suggests that the polymerization of DAS and urea had an optimal ratio, and any additional proportion was residual. Comparing the neat MUF resin with the DAS-modified MUF resin, neat MUF resin was primarily populated with free and mono-substituted urea; while the modified resin primarily contained ditri-substituted and mono-substituted urea. These results demonstrated that the incorporation of DAS facilitates urea branching of the resin system. In addition, because of the low melamine content (5% based on the weight of the liquid resin) and its addition before the acidic condensation stage, the resultant resins contained no free melamine. The effect of DAS concentration on the melamine triazine carbons cannot be determined from Table 2; however, the polymerization of DAS and melamine was verified by the NMR spectra, and their reaction was similar to that with urea (Figure 4). Furthermore, melamine exhibited a higher reactivity than that of urea.

Figure 5 presents the FTIR spectra of cured MUF resin and DAS-modified MUF resin. The characteristic peaks of DAS appear at 1000–1200 cm^−1^ and 1735 cm^−1^, and are attributed to the C–O stretching of starch and C=O stretching of the aldehyde group [27]. The peak at 1640 is considered a feature of tightly bound water in starch [26]. In the MUF resin, three typical peaks of amide are observed at 1662, 1542, and 1230 cm^−1^, which are assigned to the C=O stretching (amide I), N–H bending (amide II), and N–H in-plane and C–N stretching vibration (amide III), respectively [28]. The peak at 1383 cm^−1^ is attributed to the C–N stretching of CH_2_–N [29]; and the peak at ~1130 cm^−1^ is the C–O stretching of an aliphatic ether. The bands from 1030 to 1050 cm^−1^ were assigned to C–N or N–C–N stretching of the resin’s methylene linkage [30]. As shown in the spectra, the peak of the aldehyde group in DAS disappeared in the modified MUF resins. This indicates that DAS polymerized in situ with the MUF resin through a reaction between the aldehyde group in DAS and the amino group in urea/melamine. However, increasing the DAS concentration showed no difference in the modified MUF resins, except for a weak reappearance of the aldehyde group in the 5%DAS/MUF and 7%DAS/MUF resins. This observation suggests that DAS concentrations above 3% were excessive, which was in agreement with the ^13^C NMR analytical results.

The XRD patterns of DAS, cured MUF resin, and DAS-modified MUF resins are presented in Figure 6. DAS is a natural biomacromolecule, containing a diffuse band at 2θ = 18.2° [31]. The low crystallinity (48%) demonstrated the amorphous characteristic of DAS. MUF resin had a diffuse band at 2θ = 20.6° and a crystallinity of 54.6%. Based on the resulting patterns, it can be concluded that continually increasing concentrations of DAS gradually decreased the intensity of the diffraction band and crystallinity. This reduction is due to: (1) The polymerization of DAS reducing the mobility of the resin molecule; and (2) DAS facilitating urea branching of the resin system. These results demonstrate that DAS in situ polymerization improved the cross-linking density of the cured system, which in turn, improved the resin bond strength.

### 3.3. Curing of DAS-Modified MUF Resin

The curing of the resin is critical to the bonding substrate. Figure 7 shows the DSC curves of neat and DAS-modified MUF resins at a heating rate of 10 °C/min. The curing of MUF resin is an exothermic reaction and has a significant exothermic peak. As shown, with an increase in DAS from 1% to 7%, the peak temperature of the resin continually decreased. This is in accord with a previous study indicating that oxidized starch decreases the peak temperature of UF resin curing [17]. The decrease of temperature indicated that DAS polymerization accelerated the curing process of MUF resin. As observed in ^13^C NMR, the DAS addition resulted in an increasing hydroxylmethyl content, which could provide more reactive sites and promoted the resin system to cure easily. These results are in agreement with a recent study which showed that DAS polymerization reduces the curing enthalpy of resorcinol- formaldehyde resin [18].

### 3.4. Thermal Stability of DAS-Modified MUF Resin

Figure 8 presents the thermogravimetric and derivative thermogravimetric curves of DAS, cured MUF resin, and modified MUF resin, using DAS at different concentrations. The thermal degradation of the MUF resin is divided into four stages in accordance with the degradation peak. Stage I—the water evaporation stage—occurred from 40 to 125 °C, with a peak at ~60 °C. Stage II—the initial degradation stage—occurred from 125 to 350 °C, with an evident peak at ~275 °C. It was during this stage that small molecules and unstable chemical bonds, such as methylene ether bridges decomposing into methylene bridges [32], began to decompose. In addition, DAS also began to degrade, and showed two apparent degradation stages at 232.11 and 287.86 °C. Stage III—the skeletal structure degradation stage—occurred from 350 to 450 °C with a peak at ~404 °C. In this temperature range, the cured resin skeletal structure, including stable bonds such as methylene bridges and other cross-linking structures, began the degradation process. Stage IV—the carbon formation stage—occurred from 450 to 600 °C. In this stage, weight loss resulted from the degradation of peptide bonds into various gases, such as CO, CO_2_, and NH_3_. As shown in Figure 8, the DAS-modified MUF resins exhibited degradation behavior similar to that of the neat resin. In stages II and III, the peak intensity decreased with an increase in DAS, indicating the in situ polymerization of DAS and MUF resin.

The mass loss of the various resins samples over the first three degradation stages, as well as the residual weight (RW) at 45 °C, are listed in Table 3. The mass loss of 5% from water evaporation was similar for all resins, which demonstrates the consistency of the experimental conditions. The high value in 7%DAS/MUF might be attributed to the water absorption of redundant DAS. The mass loss in stage II decreased with an increase in DAS content, due to the continual decrease in unstable methylene ether in the modified resin, as observed in the ^13^C NMR results (Table 2). In general, the mass loss in stage III also showed the same response to DAS concentration, however, in this case, the decrease could be caused by the negative effect of DAS polymerization on methylene. With regard to the RW at 450 °C, the cured resin generally exhibited an increasing trend with increasing DAS, indicating an improvement in the thermal stability of the modified resins. These results indicate that the in situ polymerization of DAS affected the chemical structure and the number of groups, which in turn conferred enhanced thermal stability to the MUF resin.

### 3.5. Microphase Separation of DAS-Modified MUF Resin

The cured resin fracture surface micrographs are presented in Figure 9. The fracture surface of the neat MUF resin is spherical, showing typical colloidal features. The MUF resin is a dispersion of colloids with a whitened appearance, as shown in Figure 1. After curing, the MUF resin shows an aggregation of colloid particles, which ultimately aggregated into a spherical structure, and the development of a coalescence structure [25]. After introducing 1%DAS, these spherical structures tended to array linearly rather than aggregate into larger spheres. The array mode indicates that DAS polymerization affected the formation and aggregation of colloid particles, which is consistent with the liquid resin’s change in appearance (Figure 1). As the DAS concentration increased, the cured resin systems showed cross-linked and microphase-separated morphology characteristic. Moreover, the fracture surface became more rough with further DAS addition. The formation of a microphase separation structure in the MUF resin may have developed as follows: In the alkaline hydroxymethylation stage of resin synthesis, the aldehyde groups in DAS reacted with the amino groups in the urea, melamine, and hydroxymethyl groups in hydroxymethylureas and hydroxymethylmelamines. As the reaction proceeded, DAS participated in the acidic condensation of resin and thus was polymerized in situ with MUF resin molecules (as shown in Figure 4). After the curing reaction—DAS—a soft, flexible biomacromolecule (soft segment) interpenetrated into the cross-linked rigid MUF resin network (hard segment), exhibiting microphase separation (as shown in Figure 10).

### 3.6. Toughening and Enhancing of DAS-Modified MUF Resin

The mechanical properties of filter paper impregnated with neat and DAS-modified MUF resins were evaluated with tensile and impact testing. The tensile strength (Ts) and elongation at break (E) are listed in Table 4. The cured MUF resin is a brittle cross-linking network; thus, the filter paper impregnated with the neat MUF resin obtained a Ts value of 17.21 MPa and a low E value of 1.33%. However, the Ts and E values of the impregnated paper gradually increased as the DAS concentration increased, indicating that the MUF resin was significantly enhanced and toughened. The impregnated paper using modified MUF resin with 3% DAS obtained a Ts of 30.11 MPa and an E value of 2.97%, reflecting an increase of 74.96% and 48.12%, respectively, over the values obtained by the neat MUF resin. These increases in mechanical properties resulted from the in situ polymerization and microphase separation structure of DAS. The former effectively improved the cross-linking density of the cured hybrid system (Figure 6); the latter made the cured system more conducive to transfer and stress dispersal when loading external force [11].

The wet shear strength and formaldehyde emission from plywood bonded with neat and DAS-modified MUF resins are shown in Figure 11. With 3% DAS, the wet shear strength significantly improved to 1.36 MPa, reflecting a 32.04% increase over that of the neat MUF resin. This improvement is due to: (1) DAS polymerization increasing the cross-linking density of the cured system, which positively affected the bond strength of the resin; (2) DAS modification accelerating the curing of MUF resin, which facilitated the complete curing of the resin; (3) DAS polymerization-induced microphase separation structure improved the toughness of the cured resin. The DAS soft segment exhibited deformation and absorbed impact energy, thereby enhancing the DAS-modified MUF resin. Thus, in response, a flexible interface may have formed to improve the interface interaction between the resin and wood substrate, thereby enhancing the resin’s bond performance. However, the low water resistance decreased the wet shear strength of the plywood. With regard to formaldehyde emission, DAS polymerization, with urea and melamine, increased the resin’s hydroxylmethyl group concentration (Table 2) and the reverse reaction produced free formaldehyde, contributing to the formaldehyde emission from plywood. Therefore, the formaldehyde emission increased with an increase in DAS.

## 4. Conclusions

To improve the toughness and enhance the adhesive properties of the MUF resin, DAS/MUF hybrid resin systems were prepared via in situ polymerization. The aldehyde groups in DAS participated in the synthesis and curing reaction of the MUF resin with the amino and hydroxymethyl groups. ^13^C NMR demonstrated that polymerization accelerated the MUF resin curing process and increased the cross-linking density and thermal stability of the cured system. Interpenetrating DAS soft segments formed a microphase separation structure in the rigid MUF resin cross-linked network. After introducing 3%DAS, the tensile strength and elongation at break of the filter paper impregnated with MUF resin increased by 74.96% and 48.12%, and reached 30.11 MPa and 2.97%, respectively. The wet shear strength of plywood bonded with the 3%DAS/UF resin improved by 23.08% to 1.36 MPa when compared with that bonded with the neat MUF resin. All the improvements to the MUF resin are attributed to in situ polymerization and the resultant microphase separation, which are effective approaches to toughening and enhancing highly cross-linked thermosetting resins.

## Figures and Tables

**Figure 1 polymers-11-01167-f001:**
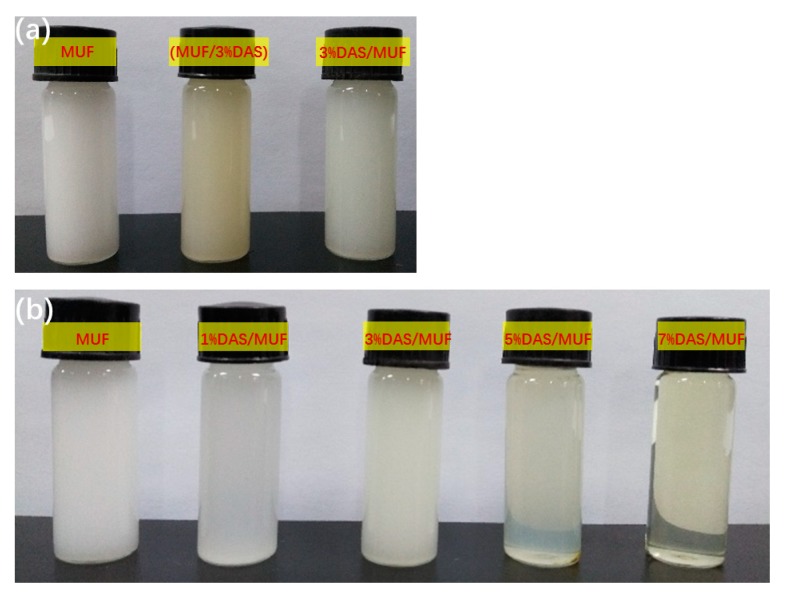
The appearances of MUF resins and DAS-modified MUF resins with different addition stages (**a**) and levels (**b**) of DAS. 3%DAS/MUF and MUF/3%DAS resin represent the DAS was added with the first and second part of urea and melamine, respectively. 3%DAS/MUF represents the DAS was added with the second urea and melamine.

**Figure 2 polymers-11-01167-f002:**
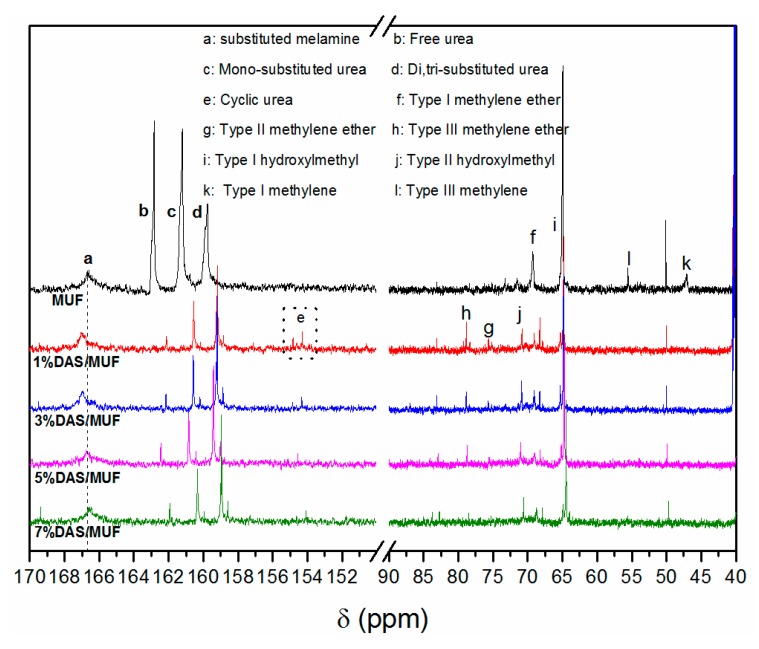
Quantitative solution ^13^C NMR spectra of neat and DAS-modified MUF resins with different levels.

**Figure 3 polymers-11-01167-f003:**
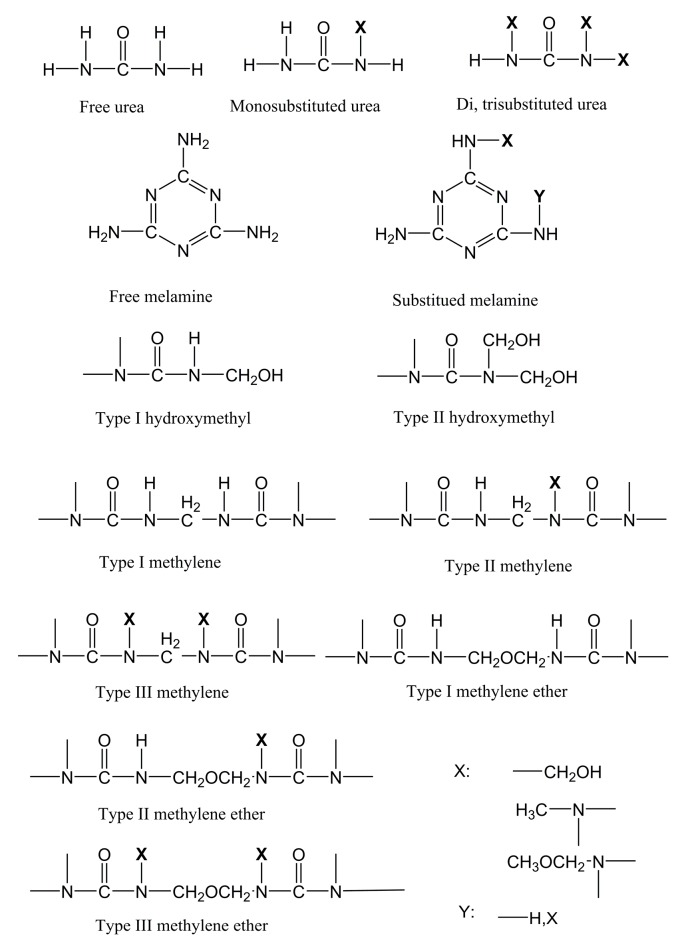
Chemical structure of various carbon groups.

**Figure 4 polymers-11-01167-f004:**
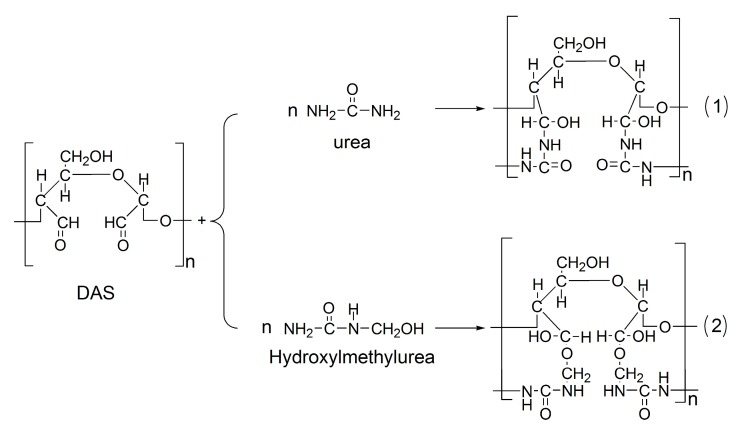
Polymerization of DAS and urea and hydroxymethylurea.

**Figure 5 polymers-11-01167-f005:**
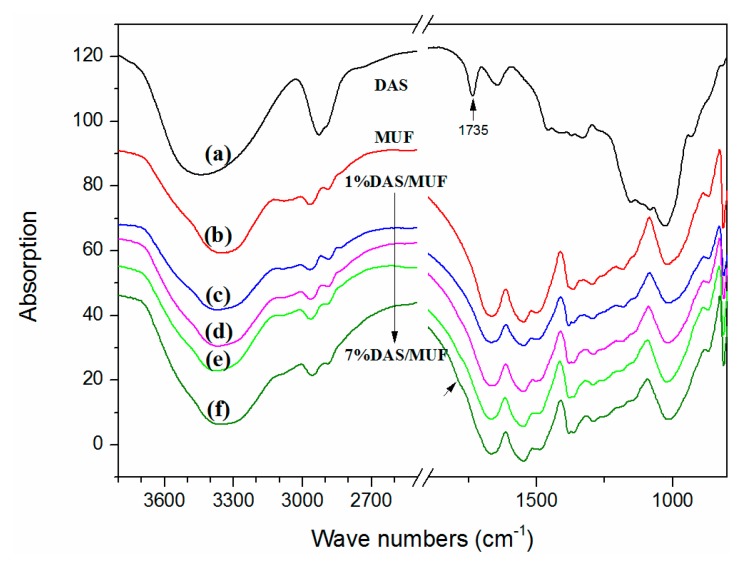
FTIR spectra of cured (**a**) MUF resin; (**b**) 1%DAS/MUF resin; (**c**) 3%DAS/MUF resin; (**d**) 5%DAS/MUF resin; and (**e**) 7%DAS/MUF resin.

**Figure 6 polymers-11-01167-f006:**
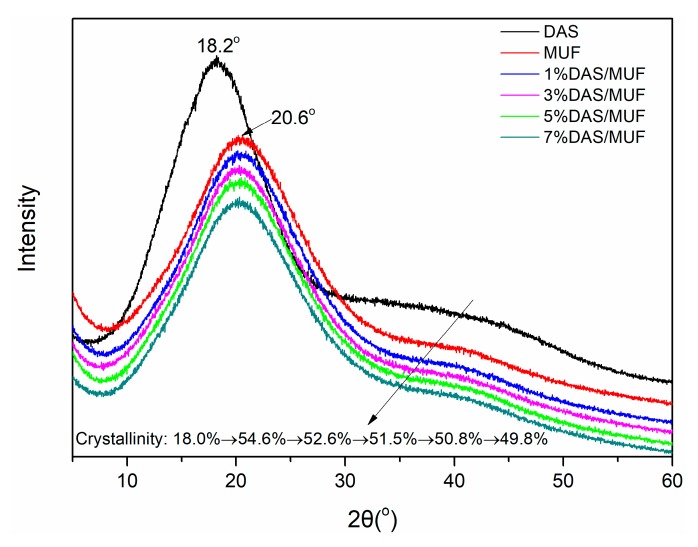
XRD patterns of DAS, the cured MUF resin and DAS-modified MUF resins with different DAS levels.

**Figure 7 polymers-11-01167-f007:**
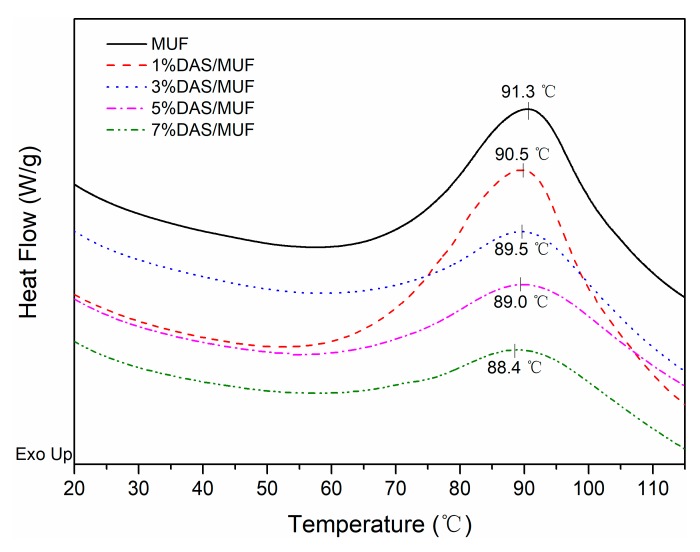
DSC curves of neat and DAS-modified MUF resins with different DAS levels.

**Figure 8 polymers-11-01167-f008:**
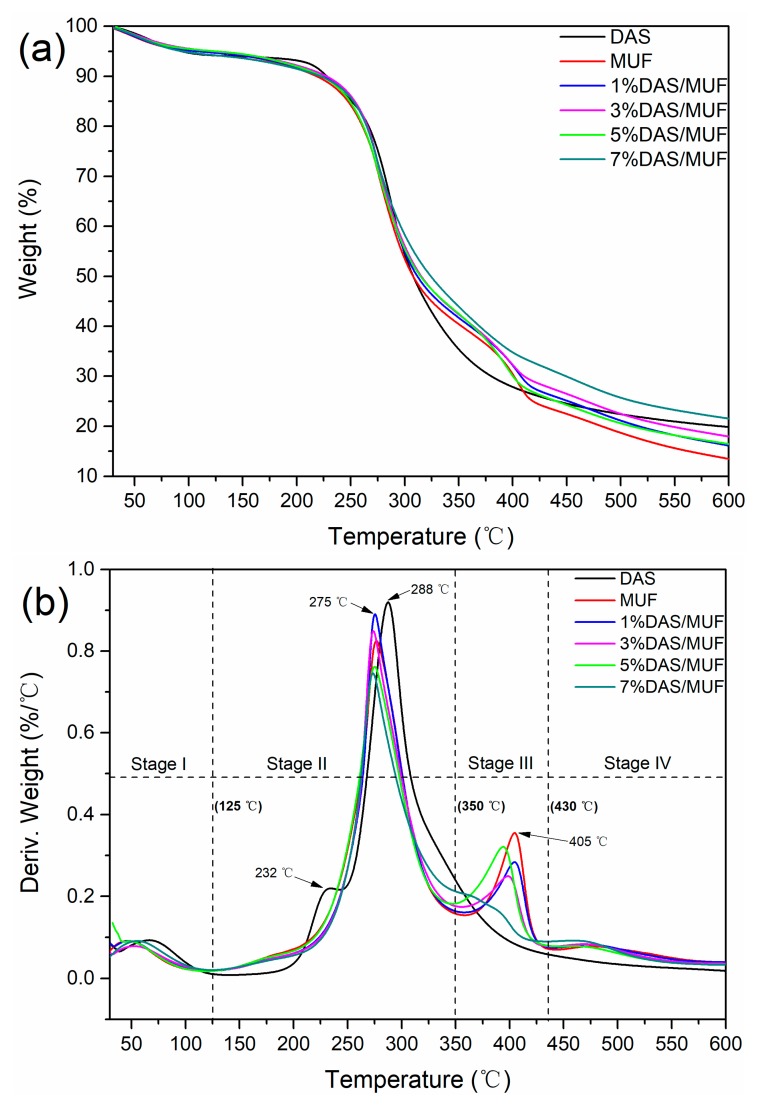
Thermogravimetric (**a**) and derivative thermogravimetric (**b**) curves of DAS, the cured MUF resin and DAS-modified MUF resins with different DAS level.

**Figure 9 polymers-11-01167-f009:**
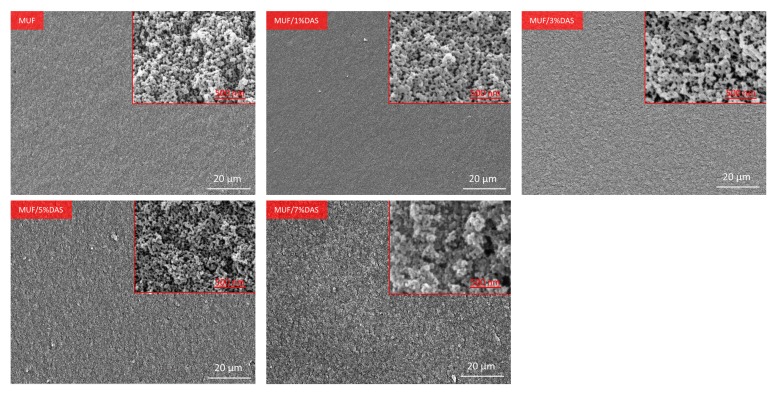
Fracture surface micrographs of the cured MUF resin and DAS-modified MUF resins with different DAS levels.

**Figure 10 polymers-11-01167-f010:**
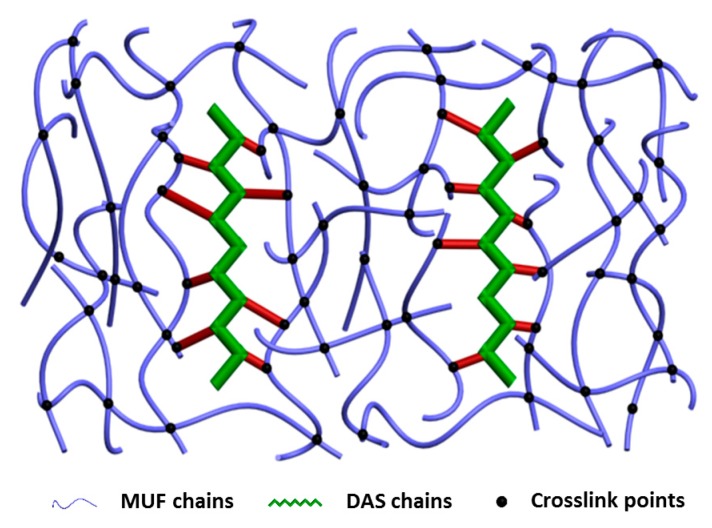
In situ polymerization and microphase separation structure in DAS-modified MUF resin.

**Figure 11 polymers-11-01167-f011:**
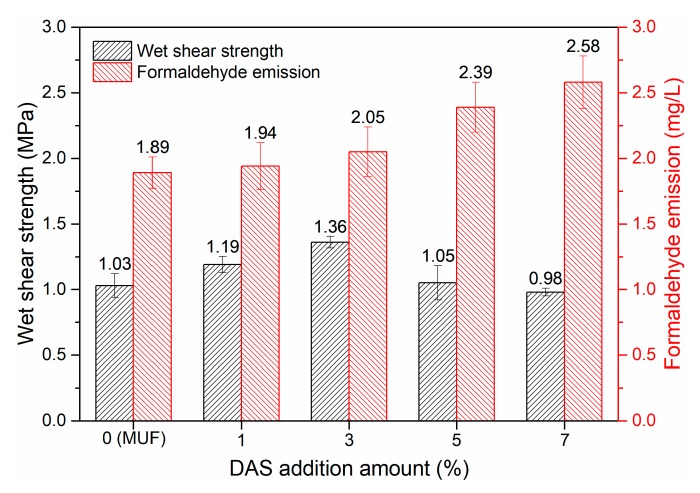
Wet shear strength of the plywood bonded by MUF resin and DAS-modified MUF resins with different DAS levels.

**Table 1 polymers-11-01167-t001:** Physical properties of neat and DAS-modified MUF resins.

Resin	Solid Content (%)	Viscosity (mPa.s)	Curing Time (s)
MUF	49.67 ± 0.16	25.9 ± 2.2	161 ± 4
1%DAS/MUF	50.24 ± 0.14	40.5 ± 2.1	155 ± 5
3%DAS/MUF	51.33 ± 0.10	53.6 ± 2.0	124 ± 2
5%DAS/MUF	52.44 ± 0.13	60.3 ± 2.7	119 ± 5
7%DAS/MUF	53.02 ± 0.15	70.6 ± 2.4	112 ± 2

**Table 2 polymers-11-01167-t002:** Percentage integration values for various methylene and carbonyl carbons of neat MUF and DAS-modified MUF resins determined from ^13^C Nuclear Magnetic Resonance (^13^C NMR) spectra.

Groups	Shift (ppm)	Resins				
		MUF (%)	1%DAS/MUF (%)	3%DAS/MUF (%)	5%DAS/MUF (%)	7%DAS/MUF (%)
**Total hydroxylmethyl**		69.50	76.98	78.56	80.57	80.98
Type I	64.3	65.67	69.89	69.81	70.09	71.29
Type II	70.6	3.83	7.09	8.75	10.48	9.69
**Total methylene**		13.08	--	--	--	--
Type I	47.1	7.27	--	--	--	--
Type II	53.9	1.45	--	--	--	--
Type III	55.6	4.36	--	--	--	--
**Total methylene ether**		16.26	20.8	19.06	18.36	17.36
Type I	69.3	13.46	11.22	13.08	12.08	13.11
Type II	75.2	1.78	3.26	2.29	0.98	0.52
Type III	78.5	1.02	6.32	3.69	4.30	3.73
free formaldehyde	82.9	1.16	2.22	2.38	2.07	1.66
**Total CH_2_**		100	100	100	100	100
Free urea	161.9	33.10	6.88	5.58	5.18	3.27
Mono-substituted urea	160.3	43.03	24.58	24.91	27.92	29.27
Di, tri-substituted urea	158.9	23.86	65.32	67.19	64.68	64.82
Cyclic urea	155	--	3.23	2.32	2.21	2.64
**Total urea**		100	100	100	100	100
Free melamine	167.8	--	--	--	--	--
Substituted melamine	166.5	100	100	100	100	100
**Total melamine**		100	100	100	100	100

**Table 3 polymers-11-01167-t003:** The mass loss (%) of different adhesive samples in three degradation stages and the residual weight (RW) at 450 °C.

Adhesive (Water/Egg White)	Mass Loss			RW (%) at 450 °C
	Stage I	Stage II	Stage III	
MUF	5.32	54.29	17.90	22.48
1%DAS/MUF	5.33	53.01	16.52	25.13
3%DAS/MUF	5.06	52.60	15.84	26.49
5%DAS/MUF	5.02	52.54	18.17	24.27
7%DAS/MUF	5.81	50.49	13.88	29.81

**Table 4 polymers-11-01167-t004:** Tensile strength (Ts) and Elongation at break (E) of the filter papers impregnated by the neat and DAS-modified MUF resins with different DAS levels.

Resin	Ts (MPa)	E (%)
MUF	17.21 ± 2.33	1.33 ± 0.42
1%DAS/MUF	22.94 ± 1.79	1.79 ± 0.33
3%DAS/MUF	30.11 ± 2.36	1.97 ± 0.56
5%DAS/MUF	29.28 ± 1.58	2.22 ± 0.54
7%DAS/MUF	31.05 ± 2.02	2.34 ± 0.61

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
