# Peer review of "Toughening and Enhancing Melamine–Urea–Formaldehyde Resin Properties via in situ Polymerization of Dialdehyde Starch and Microphase Separation"

_polymers, 2019, doi:10.3390/polym11071167_

Round 1
Reviewer 1 Report
This shows the effect of oxidised starch (OS) on the properties of MUF resins. It is worth publishing, though not entirely novel as other work on F/OS resins has been published and should be cited (DOI: 10.16865/j.cnki.1000-7555.2019.0021 and DOI: 10.15376/biores.12.4.7590-7600 )
Some points
The Kissinger model doesnt look very appropriate here as there are multiple peaks in the DSC which might skew the analysis
I couldnt really follow the discussion around lines 240 "due to the absence of acidic conditions". - but that sample was added just prior to acidification - do they mean absence of basic conditions? The others all had basic conditions and thus condensed nicely with the MUF
How is the crystallinity calculated in the XRDs? I find it hard to believe say that pure MUF resin has any crystallinity? Any broad peaks in the xrd is just an amorphous halo unless you do a lot of work proving otherwise.
The SEM explanations seem wrong. I cant see any rod like inclusions - there are some white tips in the expanded 3% photo that might be beam damage or imaging artifact, some dark areas appear to be holes, probably where the gas escaped from, and the rest is uniform grey.
The authors should answer these points, then the article would be publishable
Author Response
Dear Editors and Reviewers,
Journal: Polymers
Special issue: Thermosets Ⅱ
Thank you for your letter and the reviewers’ comments concerning our manuscript entitled “Toughening and Enhancing Melamine–urea–Formaldehyde Resin Properties via in situ Polymerization of Dialdehyde Starch and Microphase Separation” (Manuscript ID: polymers-520865). Those comments are valuable and very helpful for revising and improving our paper, as well as the important guiding significance to our researches. We have studied comments carefully and made correction that we hope to meet with approval. Revised portion are marked in red in the Main Article using the “Track Changes” function. The main corrections in the paper and the responds to the reviewer’s comments are as flowing:
Reviewer 1:
Point 1:
This shows the effect of oxidised starch (OS) on the properties of MUF resins. It is worth publishing, though not entirely novel as other work on F/OS resins has been published and should be cited (DOI: 10.16865/j.cnki.1000-7555.2019.0021 and DOI: 10.15376/biores.12.4.7590-7600)
Response 1: Thank you for the useful suggestion. These works are very valuable to support and improve our paper (Line 114-117).
Point 2:
The Kissinger model doesnt look very appropriate here as there are multiple peaks in the DSC which might skew the analysis.
Response 2: The comment is right and extremely important for the preciseness of our work. We removed the corresponding figure, reference and representation after careful consideration. Accordingly, the Figure 7 was adjusted and the discussion was revised (Line 398-406). Because the curing kinetic is not the significant section, the removing will not affect the DSC analysis result.
Point 3:
I couldnt really follow the discussion around lines 240 "due to the absence of acidic conditions". - but that sample was added just prior to acidification - do they mean absence of basic conditions? The others all had basic conditions and thus condensed nicely with the MUF.
Response 3: We are very sorry to lead to your misunderstanding. Here, we are discussing the faint yellow appearance of MUF/3%DAS resin (Figure 1 (a)) which was prepared by adding 3% DAS with the second portion of urea and melamine. As shown in the experimental procedure section (2.2.1, Line 143), the second portion of urea (U2) was added after the pH was adjusted back to 8.0. Therefore, when the DAS was added, the reaction system was alkaline condition (absence of acidic condition), resulting DAS cannot participate in the polycondensation and showing a faint yellow appearance. The relative statement has been revised.
Point 4:
How is the crystallinity calculated in the XRDs? I find it hard to believe say that pure MUF resin has any crystallinity? Any broad peaks in the xrd is just an amorphous halo unless you do a lot of work proving otherwise.
Response 4: The crystallinity was determined by deconvoluting the total area of the diffractograms, obtaining the individual contributions of the crystalline and of the amorphous regions. In this way, the crystallinity by weight was obtained by applying the least-squares fit procedure elaborated by Hindeleh and Johnson (doi: 10.1007/s10973-012-2221-x). A low molar ratio MUF resin have long been considered colloids, the ordered arrangement of colloidal particles can form the crystallization (doi:10.1002/app.23230). In fact, lot of works have been published about the crystalline features of amino resins, even the domain size of cured urea–formaldehyde resin (doi: 10.1016/j.eurpolymj.2012.10.029). Certainly, the degree of crystallinity here is semiquantitative. In our study, we just adopted the deceased crystallinity of DAS/MUF resin to testify the increase of cross-linking density in the cured system on the other side.
Point 5:
The SEM explanations seem wrong. I cant see any rod like inclusions - there are some white tips in the expanded 3% photo that might be beam damage or imaging artifact, some dark areas appear to be holes, probably where the gas escaped from, and the rest is uniform grey.
Response 5: The comment is right and important. We reorganized the figure and rewrite the representation to make it easier to observation and understanding (Figure 9 and Line 528-533).
Special thanks to you for your comments.
We appreciate for Editors/Reviewers’ warm work earnestly, and hope that the corrections will meet with approval. Once again, thank you very much for your comments and suggestions. We look forward to your information about my revised papers and thank you for your good comments.
Yours sincerely,
Janlin Luo
Reviewer 2 Report
The paper is interesting and well presented. In my opinion it deserves for acceptance.
Author Response
Dear Editors and Reviewers,
Journal: Polymers
Special issue: Thermosets Ⅱ
Thank you for your letter and the reviewers’ comments concerning our manuscript entitled “Toughening and Enhancing Melamine–urea–Formaldehyde Resin Properties via in situ Polymerization of Dialdehyde Starch and Microphase Separation” (Manuscript ID: polymers-520865). Those comments are valuable and very helpful for revising and improving our paper, as well as the important guiding significance to our researches. We have studied comments carefully and made correction that we hope to meet with approval. Revised portion are marked in red in the Main Article using the “Track Changes” function. The main corrections in the paper and the responds to the reviewer’s comments are as flowing:
Reviewer 2:
Point 1:
The paper is interesting and well presented. In my opinion it deserves for acceptance.
Response 1:
Thank you very much for your recognition of our work.
We appreciate for Editors/Reviewers’ warm work earnestly, and hope that the corrections will meet with approval. Once again, thank you very much for your comments and suggestions. We look forward to your information about my revised papers and thank you for your good comments.
Yours sincerely,
Janlin Luo
Reviewer 3 Report
The authors did a commendable job in presenting their scientific findings. In the paragraph (lines 74 - 82) there is a grammatical error and an attempt should be made to reword the sentences to make them easier to read.
I also suggest that the authors make the figures of FTIR, DSC and TGA larger to make it easier for the reader to look at the data.
Author Response
Dear Editors and Reviewers,
Journal: Polymers
Special issue: Thermosets Ⅱ
Thank you for your letter and the reviewers’ comments concerning our manuscript entitled “Toughening and Enhancing Melamine–urea–Formaldehyde Resin Properties via in situ Polymerization of Dialdehyde Starch and Microphase Separation” (Manuscript ID: polymers-520865). Those comments are valuable and very helpful for revising and improving our paper, as well as the important guiding significance to our researches. We have studied comments carefully and made correction that we hope to meet with approval. Revised portion are marked in red in the Main Article using the “Track Changes” function. The main corrections in the paper and the responds to the reviewer’s comments are as flowing:
Reviewer 3:
Point 1:
The authors did a commendable job in presenting their scientific findings. In the paragraph (lines 74 - 82) there is a grammatical error and an attempt should be made to reword the sentences to make them easier to read.
Response 1: The comment is right. We corrected our expression (Line 74-77).
Point 2:
I also suggest that the authors make the figures of FTIR, DSC and TGA larger to make it easier for the reader to look at the data.
Response 2: The comment is right and important. We resized these figures and will provided the requested size and dpi in typesetting.
Special thanks to you for your comments.
We appreciate for Editors/Reviewers’ warm work earnestly, and hope that the corrections will meet with approval. Once again, thank you very much for your comments and suggestions. We look forward to your information about my revised papers and thank you for your good comments.
Yours sincerely,
Janlin Luo